

# A bottom-up quantification of foliar mercury uptake fluxes across Europe

Lena Wohlgemuth[1,*], Stefan Osterwalder[2], Carl Joseph[1], Ansgar Kahmen[1], Günter Hoch[1], Christine Alewell[1], Martin Jiskra[1,*]

[1]Department of Environmental Sciences, University of Basel, Basel, Switzerland
[2]Institut des Géosciences de l'Environnement, Université Grenoble Alpes, CNRS, IRD, Grenoble INP, Grenoble, France

*Correspondence to: lena.wohlgemuth@unibas.ch; martin.jiskra@unibas.ch

**Abstract.** The exchange of gaseous elemental mercury, Hg(0), between the atmosphere and terrestrial surfaces remains poorly understood mainly due to difficulties in measuring net Hg(0) fluxes on the ecosystem scale. Emerging evidence suggests foliar uptake of atmospheric Hg(0) to be a major deposition pathway to terrestrial surfaces. Here, we present a bottom-up approach to calculate Hg(0) uptake fluxes to aboveground foliage by combining foliar Hg uptake rates normalized to leaf area with species-specific leaf area indices. This bottom-up approach incorporates systematic variations in crown height and needle age. We analyzed Hg content in 583 foliage samples from six tree species at 10 European forested research sites along a latitudinal gradient from Switzerland to Northern Finland over the course of the 2018 growing season. Foliar Hg concentrations increased over time in all six tree species at all sites. We found that foliar Hg uptake rates normalized to leaf area were highest at the top of the tree crown. Foliar Hg uptake rates decreased with needle age of multi-year old conifers (spruce and pine). Average species-specific foliar Hg uptake fluxes during the 2018 growing season were $18 \pm 3$ µg Hg m$^{-2}$ for beech, $26 \pm 5$ µg Hg m$^{-2}$ for oak, $4 \pm 1$ µg Hg m$^{-2}$ for pine and $11 \pm 1$ µg Hg m$^{-2}$ for spruce. For comparison, the average Hg(II) wet deposition flux measured at 5 of the 10 research sites during the same period was $2.3 \pm 0.3$ µg Hg m$^{-2}$, which was four times lower than the site-averaged foliar uptake flux of $10 \pm 3$ µg Hg m$^{-2}$. Scaling up site-specific foliar uptake rates to the forested area of Europe resulted in a total foliar Hg uptake flux of approximately $20 \pm 3$ Mg during the 2018 growing season. Considering that the same flux applies to the global land area of temperate forests, we estimate a foliar Hg uptake flux of $108 \pm 18$ Mg. Our data indicate that foliar Hg uptake is a major deposition pathway to terrestrial surfaces in Europe. The bottom up approach provides a promising method to quantify foliar Hg uptake fluxes on an ecosystem scale.



## 1 Introduction

Mercury (Hg) is a toxic pollutant ubiquitous in the environment due to long-range atmospheric transport. Anthropogenic emissions of Hg into the atmosphere mainly originate from burning of coal, artisanal and small-scale gold mining and non-ferrous metal and cement production while geogenic emission occur from volcanoes and rock weathering (UN Environment, 2019). Atmospheric Hg is deposited to terrestrial surfaces and the ocean and can be re-emitted back to the atmosphere (Bishop et al., 2020; Obrist et al., 2018). The residence time of Hg in the atmosphere and its transfer to land and ocean surfaces mainly depends on its speciation (Driscoll et al., 2013). Gaseous elemental mercury Hg(0) is the dominant form (> 90 %) of atmospheric Hg (Sprovieri et al., 2017), exhibiting a residence time of several months to more than a year (Ariya et al., 2015; Saiz-Lopez et al., 2018). Atmospheric Hg will ultimately be transferred to water and land surfaces by wet or dry deposition. In the wet deposition process, Hg(0) is oxidized in the atmosphere to water-soluble Hg(II) and washed down to the Earth surface by precipitation (Driscoll et al., 2013). Wet deposition fluxes of Hg(II) to terrestrial surfaces are well constrained and direct measurements are coordinated in regional and international atmospheric deposition monitoring programs (EMEP, NADP) (EMEP, 2016; Prestbo and Gay, 2009; Wängberg et al., 2007; Weiss-Penzias et al., 2016).

Dry deposition fluxes of Hg(0) and Hg(II) to the earth surface are less constrained owing to challenges in measuring net ecosystem exchange fluxes (Driscoll et al., 2013; Zhang et al., 2009) and atmospheric Hg(II) concentrations (Jaffe et al., 2014). The dry deposition of Hg can occur by vegetation uptake and subsequent transfer to the ground via litterfall (Risch et al., 2017; Wang et al., 2016), by wash-off from foliar surfaces via throughfall (Wright et al., 2016) or by direct deposition to terrestrial surfaces and soils (Obrist et al., 2014). Hg dry deposition is not routinely monitored by most environmental programs. Consequently, atmospheric mercury models inferring Hg dry deposition across Europe during summer months lack observational constraints (Gencarelli et al., 2015). Ecosystem scale mass balance studies, however, revealed that litterfall deposition to forest floors exceeds wet deposition (Bushey et al., 2008; Demers et al., 2007; Graydon et al., 2006; Grigal, 2002; Rea et al., 2002; Risch et al., 2012, 2017; St. Louis et al., 2001; Teixeira et al., 2012; Zhang et al., 2016). Several lines of evidence suggest that uptake of atmospheric Hg(0) by vegetation represents an important process in terrestrial Hg cycling: i) isotopic fingerprinting studies revealed that approximately 90 % of Hg in foliage and 60 % – 90 % of Hg in soils originate from atmospheric Hg(0) uptake by vegetation (Demers et al., 2013; Enrico et al., 2016; Jiskra et al., 2015; Zheng et al., 2016), ii) observations of foliar Hg concentrations increase with exposure time to atmospheric Hg(0) (Assad et al., 2016; Ericksen and Gustin, 2004; Fleck et al., 1999; Frescholtz et al., 2003; Laacouri et al., 2013; Millhollen et al., 2006; Rea et al., 2002) while Hg uptake via the root system was found to be minor (Assad et al., 2016; Frescholtz et al., 2003; Millhollen et al., 2006), iii) atmospheric Hg(0) correlates with the photosynthetic activity of vegetation suggesting that summertime minima in atmospheric Hg(0) in the Northern hemisphere are controlled by vegetation uptake (Jiskra et al., 2018; Obrist, 2007)

The exact mechanism of the atmosphere-foliar Hg(0) exchange is not yet fully understood. Laacouri et al. (2013) observed highest Hg concentrations in leaf tissues as opposed to leaf surfaces and cuticles, implying that Hg(0) diffuses into the leaves. Exposing plants to Hg(0) in form of enriched Hg isotope tracers, Rutter et al. (2011) found that plant Hg uptake was mainly to the leaf interior. Leaf Hg content correlated with stomatal density (Laacouri et al., 2013) suggesting that stomatal uptake represents the main pathway. Nonstomatal uptake was observed by Stamenkovic and Gustin (2009) under conditions of reduced stomatal aperture implying adsorption of atmospheric Hg to cuticles surfaces. Re-emission of Hg from foliage can occur by photoreduction of Hg(II) to Hg(0) and





subsequent volatilization (Graydon et al., 2006). The re-emission potential of Hg previously taken up by foliage
and strongly complexed in plant tissue (Manceau et al., 2018) was suggested to be lower than the re-emission
potential of surface-bound Hg (Jiskra et al., 2018; Yuan et al., 2019).
Hg contents in foliage were shown to be species-specific (Blackwell and Driscoll, 2015; Laacouri et al., 2013;
Navrátil et al., 2016; Obrist et al., 2012; Rasmussen et al., 1991). It is currently unresolved if deciduous broad
leaves accumulate higher Hg concentrations than needles (Blackwell and Driscoll, 2015; Navrátil et al., 2016) or
if it is the other way around (Hall and St. Louis, 2004; Obrist et al., 2011, 2012). Deciduous species shed their
leaves at the end of the growing season, whereas most conifers grow needles over multiple years and continue to
accumulate Hg, resulting in increasing Hg concentrations with needle age (Hutnik et al., 2014; Navrátil et al.,
2019; Ollerova et al., 2010). Furthermore, Hg concentrations in foliage have been shown to vary within the canopy
(Bushey et al., 2008). Physiological differences between deciduous and coniferous tree species and inconsistent
sampling of needle age and canopy height may have contributed to the uncertainty in literature whether deciduous
or coniferous species take up more Hg.
The goal of this study was to improve the understanding of foliar Hg(0) uptake and quantify foliar uptake fluxes
at European forest research sites. The objectives were to: 1) determine the temporal evolution of Hg concentrations
and the Hg pool in foliage of 6 tree species at 10 European research sites along a south-north transect from
Switzerland to Finland over the 2018 growing season, 2) investigate the effect of needle age, crown height and
tree functional group on foliar Hg uptake, 3) quantify foliar Hg uptake fluxes per m$^2$ ground surface area based
on the temporal evolution of the foliar Hg pool over the growing season. 4) estimate the foliar uptake fluxes for
Europe and temperate forests globally by scaling up species-averaged foliar uptake rates determined in this study
to respective forest areas.

**2 Materials and Methods**
**2.1 Site description**
Foliage samples were collected from 10 European research sites located along a south-north transect from
Switzerland to Scandinavia (Fig. 1). The Hölstein site in Switzerland comprises the Swiss Canopy Crane II
(SCCII) operated by the Physiological Plant Ecology Group of the University of Basel (Kahmen et al., 2019). Our
principal site Hölstein allowed to systematically access the entire canopy through the gondola of a crane. The
research sites Schauinsland and Schmücke are part of the air monitoring network of the German Federal
Environment Agency (UBA) (Schleyer et al., 2013). Hyltemossa, Norunda, Svartberget and Pallas are Integrated
Carbon Observation System (ICOS) sites operated by Lund University (LU), the Swedish University of
Agricultural Sciences (SLU) and the Finnish Meteorological Institute (FMI) (Lindroth et al., 2015, 2018; Lohila
et al., 2015). Hurdal is a prospective ICOS Ecosystem station, an ICP Forests Level II Plot and a European
Monitoring and Evaluation Programme (EMEP) air measurement site operated by the Norwegian Institute of
Bioeconomy Research (NIBIO) and the Norwegian Institute for Air Research (NILU) (Lange, 2017). Bredkälen
and Råö are Swedish EMEP air measurement sites operated by the Swedish Environmental Research Institute
(IVL) (Wängberg et al., 2016; Wängberg and Munthe, 2001). Tree species composition differed between sites.
Hölstein, for instance, is a mixed forest harbouring 14 different tree species while Hyltemossa is an exclusive





spruce stand (see Table S5 for details). At 5 locations (Schauinsland, Schmücke, Råö, Bredkälen and Pallas)
Hg(II) wet deposition measurements were performed by the operators of the research sites.

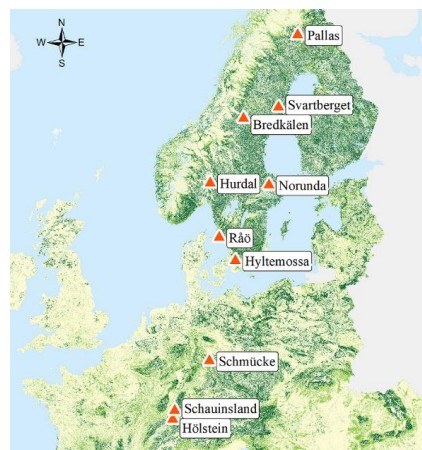


**Fig. 1: Research sites for foliage sampling during the 2018 growing season. Base map corresponds to the Joint Research**
**Centre (JRC) Pan-European Forest Type Map 2006 (JRC, 2010; Kempeneers et al., 2011). Reuse is authorized under**
**reuse policy of the European Commission (EU, 2011).**

**2.2 Sample collection**
Foliage sampling strategy was guided by the ICP Forests Programme sampling manual (Rautio et al., 2016),
requesting to take samples that have developed under open sunlight from the top third of the crown canopy. At 4
sites (Svartberget, Hyltemossa, Norunda and Hölstein) we complied with the ICP Forest sampling protocol. At 6
sites (Pallas, Bredkälen, Hurdal, Råö, Schmücke and Schauinsland) we had to adapt the sampling strategy to local
conditions and available equipment. At our focus research site in Hölstein, Switzerland a crane allowed access to
the top of the crown and vertical sampling of beech, oak and spruce. Since pine did not grow needles at ground
level we did not sample their vertical profiles. Vertical sampling of spruce needles in Hölstein during 2018 was
repeated in 2019 with five spruce trees because only two spruce trees had been sampled during 2018 of which one
died from drought induced stress by the end of the 2018 growing season (Schuldt et al., 2020). The relative effect
of height on Hg accumulation in Hölstein spruce needles is therefore investigated with data from the growing
season 2019. Samples at Hyltemossa and Svartberget were cut from tree canopies using a 20 m telescopic scissors
and at Hurdal using a 3 m telescopic scissors. At Norunda samples were shot from the tree canopies using a
shotgun. At Schauinsland, Schmücke, Råö and Bredkälen we used a 5 m telescopic scissors for cutting the
branches in the lower half of the crown. At Pallas and Råö branches were cut from low-growing trees at breast
height. We collected intact leaves at three to six time points during the 2018 growing season. Samples from at
least three different branches of the same tree were pooled to a composite sample. We sampled at least three trees
per species (one to four species) with the exception of Råö where only one oak and one spruce tree were available.
Sampling and sample preparation was conducted using clean nitryl gloves. Leaves were cut from outermost
branches. All samples were stored in Ziplock bags in the freezer until analysis. Sampling dates are reported in
Table S1 for each site. At Hölstein atmospheric Hg(0) was measured integrated over the whole sampling period
by using passive air samplers (PAS) as described by McLagan et al. (2016). PAS were exposed at ground level





(1.6 m) under the canopy at four locations on the plot and additionally at three heights of 10 m, 19 m and 35 m on
the crane railing (details in S7 and Fig. S4) from 15 May 2018 to 18 October 2018. The PAS air measurement
campaign at Hölstein was repeated in 2019 with PAS exposed at 1.5 m, 10 m, 19 m and 35 m height at the crane
from 16 May 2019 to 12 September 2019. Measurement of one of the PAS installed at 10 m height in 2019 was
excluded from further analysis because it produced an implausible high result which can probably be traced back
to a measurement error. Under dry conditions at noon time on 17 July 2019 we measured stomatal conductance
to water vapor of beech, pine and oak from the crane gondola at Hölstein using an SC-1 Leaf Porometer (Meter
Group, Inc. USA).
**2.3 Sample preparation and measurements**
In total 584 leaf samples were collected, weighted and analyzed for leaf mass per area (LMA) and subsequently
dried and grinded for Hg concentration analysis. The projected leaf area was measured using a LI3100 Area Meter
(LI-COR Biosciences USA). We performed duplicate scans of 17 % of foliage samples and obtained a mean per
cent deviation between scans and respective duplicate scans of 3 % ± 3 %. For measuring projected needle area,
we calibrated the LI3100 with rubberized wires of known length and a diameter of 1.74 ± 0.02 mm (see S4 and
Fig. S2). For the two sites Hurdal and Pallas the performance and resolution of the LI3100 was insufficient and
unrealistic results were discarded and median values from literature were used instead (see S4 for details). For the
three ICOS sites Hyltemossa, Norunda and Svartberget we obtained LMA values measured by research staff
according to ICOS protocol (Loustau et al., 2018) (Sect. S4). Foliage samples were oven-dried at 60°C for 24 h.
We did not observe any Hg losses irrespective of drying temperatures of 25°C, 60°C and 105°C (Fig. S1). A
similar result was obtained by Yang et al. (2017) for Hg in wood and by Lodenius et al. (2003) for Hg in moss.
Dried samples were weighted and homogenously grinded in an ordinary stainless steel coffee grinder. Total Hg
concentrations were measured with atomic absorption spectrophotometry using a direct mercury analyzer (DMA-
80 Hg, Heerbrugg, Switzerland). Standard Reference Materials (SRMs) used in this study were NIST-1515 apple
leaves and spruce needle sample B from the 19th ICP Forests needle/leaf interlaboratory comparison. Standard
measurement procedures included running a quality-control pre-sequence consisting of three method blanks, one
process blank (wheat flour) and three liquid primary reference standards (PRS; 50 mg of 100 ng/g NIST-3133 in
1 % BrCl). Daily performance of the instrument was assessed based on the three liquid PRS and all data were
corrected accordingly if the measured PRS were within 90 % to 110 % of the expected value. If PRS were outside
this acceptable range, the instrument was re-calibrated. Each sequence consisted of four SRMs, one process blank
consisting of commercial wheat flour and 35 samples. Sequences were rejected if one SRM value was outside of
the certified uncertainty range (NIST-1515) or 10 % of the respective target concentration (ICP Forests spruce B)
or if the absolute Hg content of the flour blank was > 0.3 ng. The average recovery for Hg during measurement
of all samples in this study was 99.9 % ± 4.0 % (mean ± sd) (n = 15) for NIST-1515 and 101.6 % ± 6.9 % (mean
± sd) (n = 40) for ICP Forests spruce B. The process blanks exhibited an average Hg content of 0.10 ng ± 0.09 ng
(mean ± sd) (n = 23). As an additional quality control, we passed the 21[st] and 22[nd] ICP Forests needle/leaf
interlaboratory comparison test 2018/2019 and 2019/2020 for Hg.
**2.4 Bottom-up calculation of foliar Hg uptake fluxes**
Foliar Hg concentration (µg Hg $g^{-1}_{d.w.}$) of each leaf/needle sample was multiplied with the respective sample leaf
mass per area (LMA; $g_{d.w.}$ $m^{-2}_{leaf}$) to obtain foliar Hg content normalized to leaf area (µg Hg $m^{-2}_{leaf}$). Foliar Hg
uptake rates (*uptakeR$_{leaf\ area}$*; µg Hg $m^{-2}_{leaf}$ month$^{-1}$) for each tree species were derived from the change in Hg

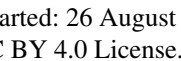



content normalized to leaf area over time (3 to 6 points in time) using a linear regression fit. Linear regression
was performed applying an ordinary least square model in the Python module statsmodels (Python 3.7.0). Linear
regression parameter ($R^2$) of each site and tree species are summarized in Table S1. Foliar Hg uptake fluxes
($uptakeF_{ground\ area}$; µg Hg $m^{-2}_{ground}$ month$^{-1}$) per ground surface area were calculated by multiplying the foliar Hg
uptake rates ($uptakeR_{leaf\ area}$) with species-specific leaf area indices (LAIs; $m^2_{leaf\ area}\ m^{-2}_{ground}$) in order to obtain
foliar Hg uptake fluxes normalized to ground surface area:
$$uptakeF_{ground\ area} = uptakeR_{leaf\ area} * LAI \tag{1}$$
Fig. *2* illustrates this flux calculation schematically. We used species-specific LAIs retrieved from a global data
base provided by Iio and Ito (2014). In total, 205 values of one-sided LAIs measured in Central Europe and
Scandinavia between a latitude of 46° N and 63° N and published in peer-reviewed journals were selected for
calculating an average LAI value of each species. Species-specific average LAI values are displayed in Table S2.
All LAI values for each species are peak-season values. To calculate the foliar uptake flux over the growing
season, the average daily uptake flux was multiplied by the length of the growing season in days. For each site,
the growing season length in days, which depends on the latitude of the site, was obtained from Garonna et al.
(2014); Rötzer and Chmielewski (2001) (Table S1). The approximate relative abundance of sampled tree species
(Table S5) at the four research sites Hölstein, Hyltemossa, Norunda and Svartberget were obtained by research
staff (pers. communication). We calculated the total foliar Hg uptake flux for these four research sites as the sum
of species-specific foliar Hg uptake fluxes of all locally dominant tree species multiplied by their relative
abundance (Table S5).

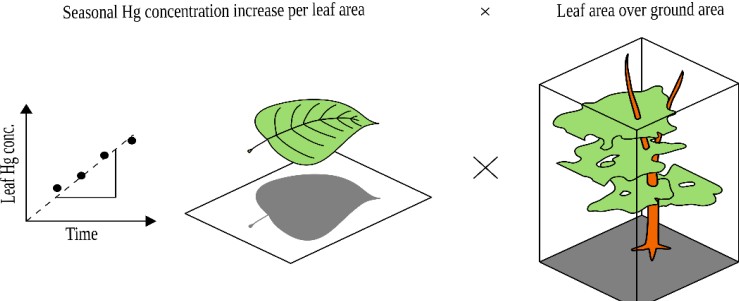


**Fig. 2: Bottom-up approach (Eq. 1) for calculating foliar Hg uptake flux per ground area (uptakeF; ng Hg $m^{-2}_{ground}$**
**month$^{-1}$). The linear regression slope of leaf Hg concentration (ng Hg $g_{d.w.}$) over time is multiplied with the respective**
**sample leaf mass per area (LMAs; $g_{d.w.}$ $m^{-2}_{leaf\ area}$). The resulting foliar Hg uptake rate per leaf area (uptakeR$_{leaf\ area}$;**
**ng Hg $m^{-2}_{leaf}$ month$^{-1}$) is then multiplied with the species-specific leaf area index (LAI; $m^2_{leaf\ area}$ $m^{-2}_{ground}$).**
**2.5 Correction factor for needle Hg uptake flux as function of needle age**
At all sites, we investigated Hg concentrations in multi-year pine and spruce needles from the current season
($y_0$, needles sprouting in spring of the sampling year) and in one-year old needles ($y_1$, needles sprouting in the year
prior to the sampling year). At 5 sites (Bredkälen, Hölstein, Hyltemossa, Schauinsland and Schmücke) we
additionally sampled two-year old ($y_2$) and three-year old ($y_3$) spruce needles. Sampling and measuring Hg uptake
in all needle age classes of a conifer tree is time-consuming and costly. In standard forest monitoring programs
young needles from age class $y_0$ or $y_1$ are usually sampled. We determined a species-specific age correction factor
($cf_{age}$) to relate the needle uptake of an entire coniferous tree to the current season ($y_0$) needles. The factor $cf_{age}$





was derived from Hg measurements of 316 needle samples of different age classes using i) the evaluated relative
Hg accumulation rate (RAR; Eq. 2), which represents the Hg accumulation of each needle age class normalized
to the Hg accumulation rate in current season ($y_0$), and ii) the respective relative biomass (RB) of each needle age
class to the total needle biomass from literature determined by Matyssek et al. (1995). Needles used to determine
the RAR were sampled by the Bavarian State Institute of Forestry at 11 ICP Forests plots in Bavaria, Germany in
2015 and 2017. Needle samples from 2015 consisted of 33 batches of spruce and 6 batches of pine samples.
Needle samples from 2017 consisted of 32 batches of spruce and 6 batches of pine samples. For spruce needles,
each batch was composed of samples of age class $y_0$ to age class $y_3$, of which 7 spruce needle batches were
composed of samples of age class $y_0$ to $y_5$ and 6 spruce needle batches of age class $y_0$ to $y_6$. For pine needles, each
batch of the two sampling years 2015 and 2017 was composed of samples of age class $y_0$ to $y_1$ and one pine needle
batch was additionally composed of samples of age class $y_2$. The RAR of spruce and pine samples of different
needle years ($y_i$, i = 1, 2, …, n) in each sample batch of the sampling years 2015 and 2017 was calculated as
follows:
$$RAR_{y_i} = \frac{c_{Hg}(y_i) - c_{Hg}(y_{i-1})}{c_{Hg}(y_0)}$$            (2)
Resulting average RARs of the spruce and pine needle samples together with the RB are presented in Table S3.
For each needle age class the factor $cf_{age}$ calculates as
$$cf_{age} = 1 * RB_{y_0} + RAR_{y_1} * RB_{y_1} + \ldots + RAR_{y_n} * RB_{y_n}$$            (3)
In accordance to our bottom-up approach for calculating the foliar Hg uptake flux (Eq. 1) the modified flux
calculation for conifers is:
$$uptakeF_{ground\ area} = cf_{age} * uptakeR_{y0;\ needle\ area} * LAI$$            (4)
Final values of $cf_{age}$ are summarized in Sect. S6, Table S3.
**2.6 Correction factor for foliar Hg uptake flux as function of crown height**
Standard foliage sampling in forest monitoring programs is from the top third of the crown (Rautio et al., 2016).
We determined a species-specific height correction factor ($cf_{height}$) allowing to scale up the treetop foliar Hg uptake
flux to whole-tree foliage. The species-specific height correction factor equals the multiplication of two ratios: i)
the ratio $r_{conc.coeff.}$ of the linear regression coefficient (ng Hg g$^{-1}_{d.w.}$ month$^{-1}$) of Hg concentrations in foliar samples
over the growing season at ground/mid canopy level to the equivalent coefficient at top canopy level and ii) the
ratio $r_{LMA}$ of average LMA at ground/mid canopy level to the average LMA at top canopy level (Eq 5).
$$cf_{height} = r_{conc.\ coeff.} * r_{LMA} = \frac{conc.\ coeff._{ground}}{conc.\ coeff._{top\ canopy}} * \frac{LMA_{ground}}{LMA_{top\ canopy}}$$            (5)
According to ecosystem models on light attenuation and photosynthesis in tree canopies (Hirose, 2004; Körner,
2013; Monsi and Saeki, 2004) the 3 top canopy layers of leaf area intercept almost 90 % of available sunlight
leaving the lower leaf layers with reduced light. We thus assume that the top 3 canopy layers of leaf area index
(LAI; m$^2_{leaf\ area}$ m$^{-2}_{ground}$) mainly consist of sun-adapted foliage (i.e. sun-leaves) with Hg uptake rates corresponding
to the uptake rates measured at top canopy. Leaf area indices and vertical foliar biomass distribution differ between
tree species (Fichtner et al., 2013; Hakkila, 1991; Sharma et al., 2016; Tahvanainen and Forss, 2008; Temesgen
et al., 2005). We did not apply a height correction for tree species with a LAI ≤ 3. For tree species with leaf area



indices > 3 we assumed the following species-specific foliar Hg uptake flux of the whole tree foliage (uptakeF)
in extension of Eq. (1):
$uptakeF_{ground\ area}\ [ng\ Hg\ m_{ground}^{-2}\ month^{-1}] = uptakeR_{top\ canopy;\ leaf\ area} * \left(3 + cf_{height} * (LAI - 3)\right)$

249         (6)

Final values of $cf_{height}$ are summarized in Sect. S9, Table S4.
**3 Results and Discussion**
**3.1 Effect of needle age on foliar Hg uptake**
Spruce and pine revealed increasing Hg concentration with needle age at all sites (Fig. S5). In order to demonstrate
the increase in Hg concentration with needle age class, we display results from Hölstein, Hyltemossa and
Schauinsland (Fig. 3). The average late season Hg concentration in one-year old ($y_1$) spruce needles was by a
factor of $1.8 \pm 0.4$ (mean $\pm$ sd between all sites) times higher than the average late season Hg concentration in
current season ($y_0$) spruce needles. From spruce needle age class $y_2$ to $y_1$ the ratio of average Hg concentrations
was $1.3 \pm 0.1$ and from $y_3$ to $y_2$ $1.4 \pm 0.1$. For pine the corresponding ratio was $1.9 \pm 0.2$ (mean $\pm$ sd between all
sites) from $y_1$ to $y_0$ needles. Consequently, needle Hg concentrations in spruce and pine almost doubled from the
season of sprouting to the subsequent growing season one year later. Needles older than one year ($y_2$, $y_3$) continue
to accumulate Hg albeit at a slower rate than younger needles ($y_0$, $y_1$). This finding is in agreement with previous
studies that reported positive trends of Hg concentration in spruce needles from age class $y_1$ to $y_4$ (Hutnik et al.,
2014; Navrátil et al., 2019; Ollerova et al., 2010).

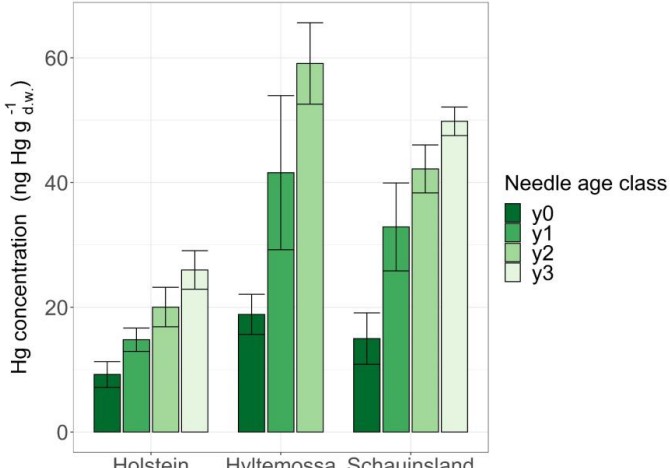


**Fig. 3: Hg concentrations (ng g⁻¹ d.w.) in spruce needles of four different age classes sampled at 3 research sites**
**(Hölstein, Hyltemossa and Schauinsland) at the end of the 2018 growing season (October – November). Age class $y_0$**
**represents current season needles, age classes $y_1$, $y_2$ and $y_3$ one-, two- and three-year old needles, respectively. Error**
**bars denote one standard deviation of samples taken from multiple trees at each site.**
We systematically investigated age dependency of Hg accumulation rates using 292 spruce and 24 pine needle
samples of age class $y_0$ to $y_6$ sampled by the Bavarian State Institute of Forestry in 2015 and 2017 (Sect. 2.5). The
relative accumulation rate (RAR) represents the Hg accumulation of an individual needle age class normalized to
the respective Hg accumulation rate in the current season $y_0$ needles (Eq. 2). Needles of all age classes continue



to accumulate Hg, which is in concurrence with our 2018 Hg concentrations of needles $y_0$ to $y_3$ (Fig. 3). However,
RAR decrease with needle age (Fig. 4). Assuming a linear decline in Hg uptake with spruce needle age, the mature
needles ($y_n$) took up -0.17 ± 0.03 (linear regression coefficient ± se) in 2015 and -0.10 ± 0.02 (linear regression
coefficient ± se) in 2017 than the previous age class $y_{n-1}$. The negative linear trend of pine needle Hg uptake was
-0.18 ± 0.02 (linear regression coefficient ± se) in 2015 samples (from $y_0$ to $y_2$ Hg uptake) and -0.17 (linear
regression coefficient) in 2017 samples (from $y_0$ to $y_1$ Hg uptake).

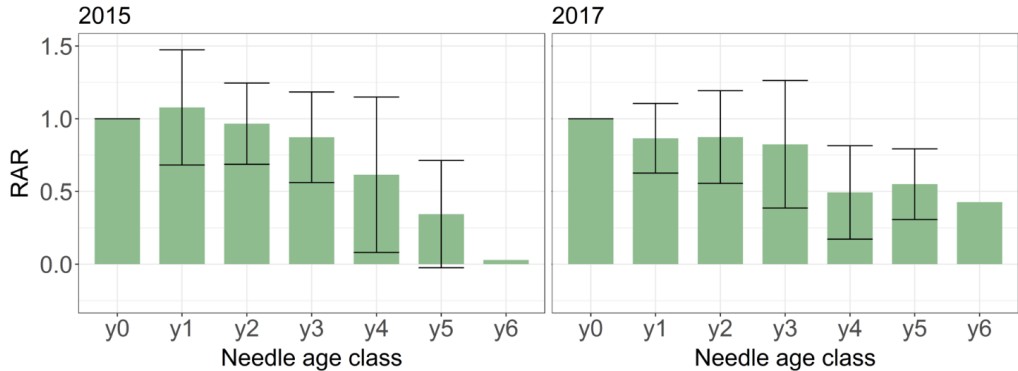


**Fig. 4: Average relative Hg accumulation rates (RAR) of 292 spruce needle samples of age class $y_0$ and $y_6$ taken by the**
**Bavarian State Institute of Forestry in the two sampling years 2015 (left) and 2017 (right). The RAR represents the**
**ratio of average Hg accumulation rate of the respective needle age class to the Hg accumulation of needle age class 0**
**($y_0$). Error bars denote one standard deviation between RAR of needles sampled from multiple trees and sites.**
The decline of Hg RAR with age could be related to a decrease in physiological activity with needle age. The rate
of photosynthesis and stomatal conductance decreases in older needles (Freeland, 1952; Jensen et al., 2015; Op
de Beeck et al., 2010; Robakowski and Bielinis, 2017; Warren, 2006; Wieser and Tausz, 2007). Consequently, a
physiologically less active older needle accumulates less Hg(0). Additionally, adsorption of Hg(0) to needle wax
layers as a possible nonstomatal uptake pathway might be minimized in older needles because ageing needles
suffer from cuticular wax degradation (Burkhardt and Pariyar, 2014; Güney et al., 2016). As older needles
exhibited higher Hg concentrations than younger needles, the Hg re-emission flux might increase with age.
Differences of Hg RARs between sampling years 2015 and 2017 (Fig. 4) could be the result of climatic conditions
during the two years like precipitation rates, temperature or vapor pressure deficit which impacts needle stomatal
conductance and possibly stomatal Hg(0) uptake (Blackwell et al., 2014).
From RAR values of our systematic needle analysis (Fig. 4) we calculated needle age correction factors ($cf_{age}$)
according to Eq. (3) in order to scale up Hg uptake fluxes determined for $y_0$ needles to Hg uptake fluxes in needles
of all age classes (Eq. 4). The correction factor $cf_{age}$ was 0.79 ± 0.03 (factor according to Eq. 3 ± se) for spruce
and 0.87 ± 0.06 (factor according to Eq. 3 ± se) for pine (see S6 for details).
**3.2 Effect of crown height on foliar Hg content**
Foliar Hg concentration, leaf mass per area (LMA) and Hg content normalized to leaf area measured at Hölstein
exhibited vertical variation with crown height (Fig. 5). In the following, we discuss all data relative to values
measured at top canopy. Top canopy represents the foliage sampling height at the sun-exposed treetop, mid canopy
describes the middle height range of sampled trees and ground level represents chest height (1.5 m).



Hg concentrations of beech (Fig. 5a), oak and spruce were lower in top canopy foliage than in foliage growing at
ground level. By the end of the growing season (October), average Hg concentration in top canopy (33 – 38 m)
beech leaves was 0.84 times and 0.72 times the average Hg concentration at mid canopy (18 – 21 m) and ground
level (1.5 m) respectively. For oak, the ratio of average Hg concentrations in top canopy (28 – 38 m) leaves to
mid canopy (19 – 22 m) leaves was 0.92 and for current season spruce needles the respective ratio was 0.85 from
top canopy (43 - 47 m) to mid canopy (25 - 34 m) needles (spruce needles sampled in September 2019, see 2.2).
LMA of foliage samples from top canopies was higher than LMA of foliage samples from lower tree heights (Fig.
5b exemplary for beech). The season-averaged LMA ratio of top canopy foliar samples to ground foliar samples
was 2.9 for beech, 1.3 for oak and 1.6 for spruce.
Because of the large vertical LMA gradient, foliar Hg content normalized to leaf area exhibited an opposite
vertical gradient with tree height compared to Hg concentrations (Fig. 5c exemplary for beech). By the end of the
growing season Hg content normalized to leaf area in top canopy (33 – 38 m) beech leaves was 1.17 times the Hg
content per area in mid canopy (18 – 21 m) and 1.91 times in ground level (1.5 m) leaves. The equivalent ratio of
Hg content per area in oak leaves was 1.13 from top canopy (28 – 38 m) to mid canopy (19 – 22 m) and 1.55 for
spruce needles from top (43 - 47 m) to mid canopy (25 - 34 m).

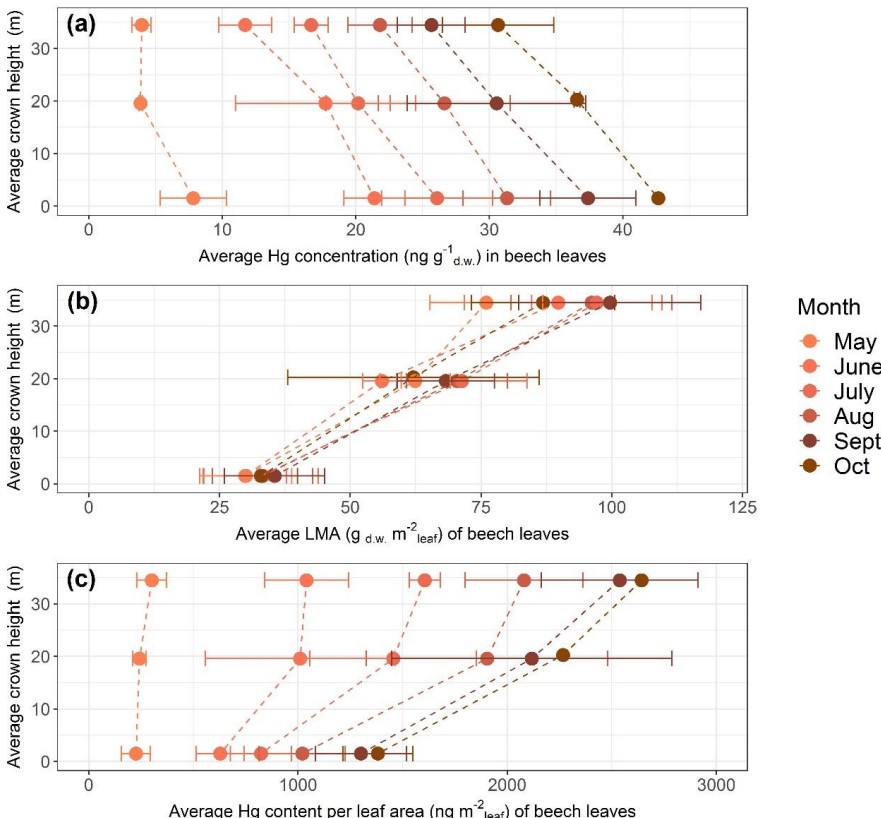


**Fig. 5: Average values of beech leaf parameters as a function of average tree crown height in meters above ground level**
**at Hölstein, Switzerland over the course of the 2018 growing season: a) Hg concentrations (ng Hg g$^{-1}$$_{d.w.}$), b) leaf mass**
**per area (LMA; g$^{1}$ m$^{-2}$$_{leaf}$ d.w.), c) Hg content normalized to projected leaf area (ng Hg m$^{-2}$$_{leaf}$). Error bars denote one**
**standard deviation of leaf samples from multiple beech trees (n = 3 – 5).**





Gradients of LMA with tree height are a result from leaf adaptation to changing light conditions and have
previously been reported by multiple studies (Konôpka et al., 2016; Marshall and Monserud, 2003; Merilo et al.,
2009; Morecroft and Roberts, 1999; Stancioiu and O'Hara, 2006; Xiao et al., 2006). Leaves exposed to intense
sunlight in tree canopies tend to grow thicker and denser thereby accumulate more photosynthesizing biomass per
unit surface area (Niinemets et al., 2001; Sonnewald, 2013). It is thus likely that foliar Hg content per gram dry
weight is diluted in sun exposed canopy leaves relative to lower growing shade leaves explaining the observed
gradient in foliar Hg concentrations with tree height (Fig. 5a). Foliar Hg content normalized to leaf area (ng Hg
$m^{-2}_{leaf}$; Fig. 5c) is derived from the multiplication of Hg concentrations and respective LMA. As the gradient of
LMA values with height (Fig. 5b) is reversed to and steeper than the gradient in Hg concentrations with height
(Fig. 5a), foliar Hg content per leaf area (Fig. 5c) decreases from top to ground level. Therefore, care has to be
taken when comparing different data sets of foliar Hg concentrations as foliar Hg concentrations depend on leaf
morphology which varies with height and between tree species.
**3.3 Effect of crown height on foliar Hg uptake rates per leaf area**
Hg uptake rates per leaf area (upkate$R_{leaf\ area}$) were higher in top canopy compared to mid canopy/ground level by
a ratio of 2.19 for beech, 1.22 for oak and 1.72 for spruce. Thus, foliage takes up more Hg per area at top canopy
level than at ground level (Fig. 6a exemplary for beech). We propose two mechanisms that possibly explain
increasing Hg uptake rates per leaf area with crown height: **(1) Vertical variation in stomatal density and**
**stomatal conductance:** Leaves from the top of the canopy (sun leaves) have been reported to exhibit a
significantly higher mean stomatal density than leaves within the canopy (shade leaves) (Poole et al., 1996). A
higher stomatal density (number of stomata pores per unit leaf area) is associated with a higher Hg content per
leaf area (Laacouri et al., 2013). The observed gradient of higher Hg uptake per leaf area towards the top canopy
(Fig. 6a) possibly reflects higher stomatal density in sun leaves compared to shade leaves at ground level.
Supplementary to stomatal density, we hypothesize that stomatal conductance to water vapor is a defining
parameter for foliar Hg uptake per area. We measured stomatal conductance under dry conditions at Hölstein at
noon on 17 July 2019 and observed higher average values in top canopy beech leaves than in ground level beech
leaves (Fig. 6b). Stomatal conductance to water vapor is subject to temporal change depending on meteorological
conditions and soil moisture content (Körner, 2013; Schulze, 1986). Nevertheless, the observed gradient in
stomatal conductance with tree height (Fig. 6b) conceivably indicates that foliar-atmosphere exchange of water
vapor and Hg(0) are related. (**2) Vertical air Hg(0) gradient:** We observed a small gradient in atmospheric Hg(0)
from 1.6 ng $m^{-3}$ at the top (35 m a.g.l) to $1.4 \pm 0.08$ ng $m^{-3}$ at ground level (1.6 m a.g.l.) integrated over the growing
season 2018 (May – October) and from 1.7 ng $m^{-3}$ (35 m a.g.l) to 1.4 ng $m^{-3}$ (1.6 m a.g.l.) integrated over the
growing season 2019 (May – September) (Fig. 6c). We hypothesize that depletion in atmospheric Hg(0) within
the canopy was driven by foliar uptake of atmospheric Hg(0) (Fu et al., 2016; Jiskra et al., 2019). The vertical
Hg(0) gradient in air possibly contributed to the gradient of Hg content per leaf area in beech, oak and spruce
from top canopy to ground/mid canopy because ground level leaf area intercepts less air Hg(0) than canopy leaf
area. A caveat to consider is that the Hg(0) concentration gradient measured depends on sampling rates of
deployed passive samplers, which were considered to be constant with height (detailed discussion in S7).


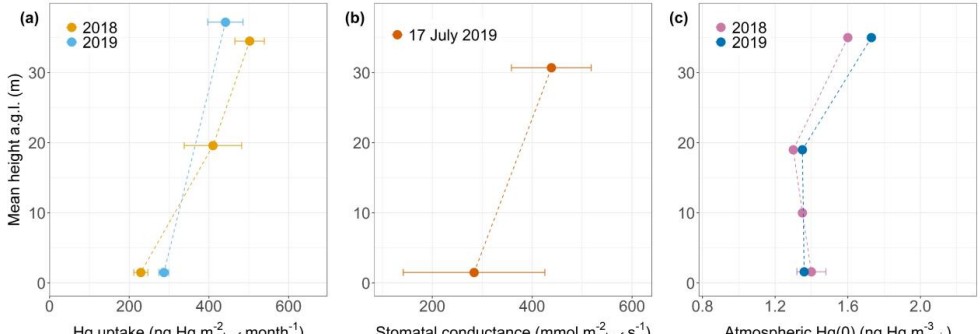


**Fig. 6: (a) foliar Hg uptake rate per leaf area (ng Hg m$^{-2}_{leaf}$ month$^{-1}$; linear regression coefficient ± se) by beech leaves at various tree heights (m) at Hölstein during two growing seasons 2018 and 2019; (b) Stomatal conductance to water vapor (mmol m$^{-2}_{leaf}$ s$^{-1}$; mean ± sd) measured in Hölstein beech leaves at top canopy and ground level under dry conditions at noon on 17 July 2019; (c) Atmospheric Hg(0) (ng Hg m$^{-3}_{air}$) at various heights in Hölstein measured with passive air samplers and integrated over the 2018 and 2019 growing season respectively. Error bars at ground level height (1.6 m) of 2018 data denote one standard deviation between 4 passive samplers.**

Re-emission of Hg(0) from foliage driven by photoreduction of Hg(II) to Hg(0) can counterbalance gross uptake
of Hg(0) (Yuan et al., 2019). Re-emission rates will be enhanced in the top of the canopy due to higher light
availability. However, re-emission rates were not large enough to compensate for higher Hg uptake per leaf area
by top canopy leaves compared to ground level leaves (Fig. 6a).

### 3.4 Effect of tree functional group (deciduous vs. conifer) on foliar Hg uptake

Broad leaves of deciduous species (beech and oak) in Hölstein exhibited on average approximately five times
higher Hg concentration increases (5.3 ± 0.6 ng Hg g$^{-1}$ d.w. month$^{-1}$; mean ± se) compared to current-season pine
and spruce needles (mean: 1.1 ± 0.4 ng Hg g$^{-1}$ d.w. month$^{-1}$; mean ± se) (Fig. 7a). Higher Hg concentrations in
broad leaves directly compared to conifer needles were also found by Blackwell and Driscoll (2015); Navrátil et
al. (2016) but not by Hall and St. Louis (2004); Obrist et al. (2011, 2012). Foliar Hg uptake rates normalized to
leaf area in Hölstein were approximately 3 times higher in broad leaves (622 ± 84 ng Hg m$^{-2}_{leaf}$ month$^{-1}$; mean ±
se) than in conifer needles (222 ± 81 ng Hg m$^{-2}_{leaf}$ month$^{-1}$; mean ± se) (Fig. 7b). Thus, our results exhibit higher
foliar Hg uptake per leaf area in broad leaves than in current-season conifer needles.

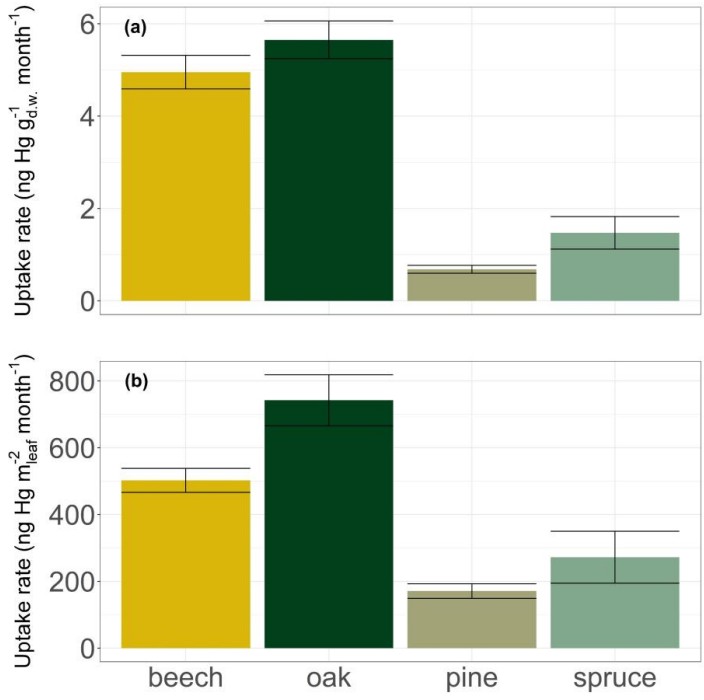

**Fig. 7: Uptake rates by leaves and current-season needles of 4 tree species at Hölstein (a) of ng Hg g⁻¹ foliage dry weight and month; (b) of Hg uptake rate normalized to leaf area in ng Hg m⁻² month⁻¹. Error bars denote standard errors of the linear regression of foliar Hg concentrations over the growing season.**

We propose that Hg uptake rates have to be assessed in the context of different physiological characteristics of conifer needles and broad leaves. Needles generally have a larger LMA ($245 \pm 62$ g m⁻² in Hölstein) than broad leaves ($79 \pm 38$ g m⁻² in Hölstein). Plant tissues with large LMA such as needles are associated with low metabolic activity including photosynthesis and respiration (Körner, 2013; Reich et al., 1997; Wright et al., 2004). Accordingly, the stomatal conductance to water vapor of canopy foliage in Hölstein on 17 July 2019 was lower for coniferous pine needles ($289 \pm 137$ mmol m⁻² s⁻¹; mean $\pm$ sd; n = 14) than for broad leaves of beech ($438 \pm 80$ mmol m⁻² s⁻¹; mean $\pm$ sd; n = 14) and oak ($849 \pm 221$ mmol m⁻² s⁻¹; mean $\pm$ sd; n = 15). The variation between foliage functional groups (conifer needles vs. broad leaves) indicates that foliar Hg uptake is related to stomatal conductance.

### 3.5 Foliar Hg uptake fluxes per ground area

We calculated foliar Hg uptake fluxes per ground area (m²$_{ground}$) by multiplying foliar Hg uptake rates per leaf area (m²$_{leaf}$) with species-specific LAI (Eq. 1). LAI values (mean $\pm$ sd) differed between tree species and were highest in spruce ($7.3 \pm 2.1$) and beech ($7.0 \pm 1.6$) and lowest in pine ($2.9 \pm 1.4$) and birch ($2.6 \pm 1.2$) (Table S2). In general, forests consisting of spruce trees with high LAI might therefore exhibit higher Hg uptake fluxes than deciduous forests with low average LAI even though Hg uptake rates per leaf area might be lower for conifer needles than for broad leaves (Sect. 3.4). We applied correction factors for needle age for conifer samples (Eq. 4) and crown height for sites where we collected top canopy samples (Hölstein, Hyltemossa, Norunda and Svartberget) (Eq. 6). The foliar Hg uptake flux showed a large variation ranging from 2 µg Hg m⁻² (Pallas, pine) to 26 µg Hg m⁻² (Schauinsland, beech) over the 2018 growing season (Fig. S6). The 4 sites where samples were

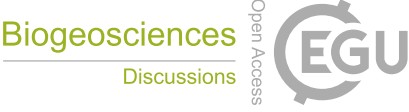



collected from top canopy exhibited a smaller range for spruce between sites from 7 to 15 µg Hg m$^{-2}$ season$^{-1}$
(Fig. 8). Given the systematic variation of Hg uptake rates with tree height (Fig. 5) we cannot exclude that the
inconsistent sampling strategy might have influenced the observed Hg uptake fluxes among the 10 sampling sites.
We will therefore not further discuss the observed variation between sites. To scale up site-based Hg uptake fluxes,
we only consider sites where we consistently sampled the top third of the canopy (Hölstein, Hyltemossa, Norunda
and Svartberget). The average foliar Hg uptake fluxes of each species at the four crown sampling sites (mean ±
se between sites) during the 2018 growing season was 18 ± 3 µg Hg m$^{-2}$ for beech, 26 ± 5 µg Hg m$^{-2}$ for oak, 4 ±
1 µg Hg m$^{-2}$ for pine and 11 ± 1 µg Hg m$^{-2}$ for spruce (see S13 for standard errors of fluxes). Deciduous trees
exhibited higher foliar uptake fluxes compared to coniferous trees resulting from generally higher uptake rates
per leaf area (Fig. 7b) owing to higher physiological activity of deciduous trees.

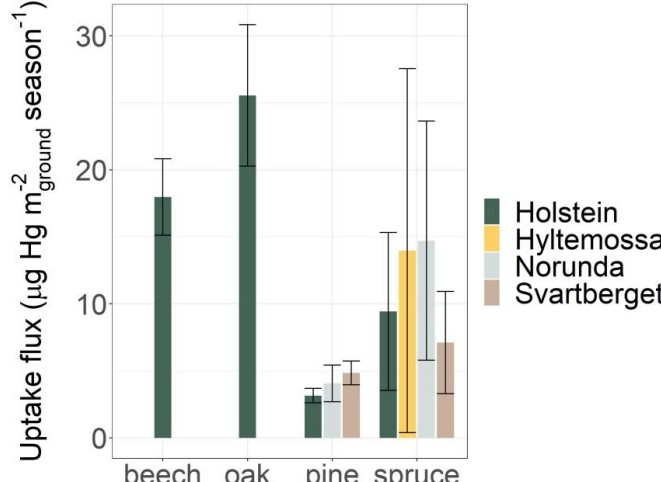


**Fig. 8: Foliar Hg uptake fluxes (µg Hg m$^{-2}$ during the 2018 growing season) at four forested research sites where foliage**
**samples were taken from crown height. Error bars indicate one standard error of the regression slope.**

**3.6 Foliar Hg uptake fluxes along a latitudinal gradient in Europe**
We calculated total Hg uptake fluxes at each research site as the sum of Hg uptake fluxes of each tree species and
research site weighted by the relative abundance of the respective tree species to the other examined tree species
at each site (Fig. 9). The average foliar Hg uptake flux of the 4 research sites where foliage samples were obtained
from tree crown heights over the 2018 growing season was 11 ± 3 µg Hg m$^{-2}$ (mean ± sd). Spruce needle Hg
uptake fluxes did not exhibit a clear trend with latitude (Fig. 8b with sites sorted for latitude).
The aboveground foliar Hg uptake fluxes per site (range 6 - 14 µg Hg m$^{-2}$ growing season$^{-1}$) are in the lower range
of published Hg litterfall fluxes in Europe and North America measured for various years, which range from 9.7
to 28.5 µg Hg m$^{-2}$ y$^{-1}$ (Demers et al., 2007; Juillerat et al., 2012; Navrátil et al., 2016; Rea et al., 1996, 2002; Risch
et al., 2012, 2017).
The average wet Hg(II) deposition fluxes measured at Schauinsland, Schmücke, Råö, Bredkälen and Pallas over
the course of the sampling period was 2.3 ± 0.3 µg Hg m$^{-2}$ (mean ± sd). Wet Hg deposition fluxes were consistently



lower than foliar Hg uptake fluxes. Our data constrain that foliar Hg uptake is a major deposition pathway to
terrestrial surfaces in Europe, exceeding direct wet deposition of Hg(II) by a factor of four. Note that this
assessment only compares Hg(0) uptake by foliage and does not take into account Hg incorporated into wood
biomass (Navrátil et al., 2019) or Hg(0) adsorbed to leaf surfaces that is washed off between sampling events as
throughfall (Demers et al., 2007; Rea et al., 1996, 2001). Total Hg(0) deposition fluxes to terrestrial ecosystems,
which also include Hg(0) deposition to soils and litter (Obrist et al., 2014, 2017; Pokharel and Obrist, 2011; Zhang
et al., 2009) are therefore expected to be higher than foliar uptake fluxes quantified here.

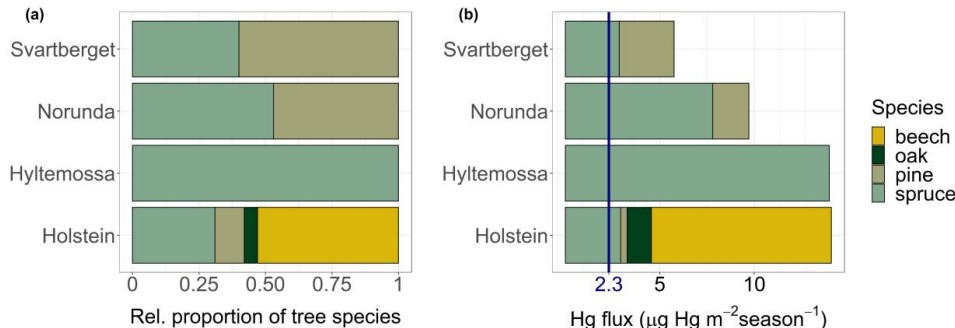


**Fig. 9: (a) Relative proportion of tree species to each other and (b) foliar Hg fluxes (µg Hg m$^{-2}$ over the 2018 growing**
**season) at 4 European research sites ordered by latitude from South (Hölstein at 47° N) to North (Svartberget at 64°**
**N); blue label of 2.3 µg Hg m$^{-2}$ season$^{-1}$ corresponds to the average wet deposition flux measured at 5 sites over the**
**course of the sampling period**

Averaging species-specific foliar Hg uptake fluxes and weighting them with the tree species proportion in Europe
derived from Brus et al. (2012) yields an average foliar Hg uptake flux for Europe of 10.4 ± 2 µg Hg m$^{-2}$ over the
2018 growing season (weighted mean ± se). Extrapolation of this weighted mean to the land area of European
forests (192.672 × 10$^6$ hectares) results in a foliar flux of 20 ± 3 Mg Hg during the 2018 growing season (see
Sect. S12 for details on flux extrapolation and Sect. S13 for error propagation). Under the assumption that tree
species in the global temperate zone are distributed equally to tree species in Europe we estimated an approximate
foliar flux of 108 ± 18 Mg Hg to the area of global temperate forests (1.04 * 10$^9$ hectares) (Tyrrell et al., 2012)
during the 2018 growing season. This global extrapolation is at the lower end of global Hg litterfall deposition
flux (163 Mg Hg yr$^{-1}$) estimated for temperate forests based on a Hg litterfall flux database of measurements
between 1995 – 2015 (Wang et al., 2016). In order to obtain a more precise foliar Hg uptake flux estimate to
European and global forests, improved spatially resolved foliar Hg data and comprehensive ground-based forest
statistics of tree species composition are needed.
**4. Conclusion**
We observed that Hg concentrations in foliage increased over the growing season in broadleaf and coniferous
trees. Concentrations of Hg in multi-year needles increased with age. The foliar Hg uptake normalized to leaf area
was higher on top of the canopy than at ground level. The temporal and vertical variation of foliar Hg uptake
fluxes are consistent with the notion that stomatal uptake represents the main deposition pathway to atmospheric
Hg(0). We emphasize that standardized sampling strategies and reporting of sampling height and needle age class
is essential to allow for comparison of foliar Hg results between different studies.





We developed a bottom-up approach to quantify foliar Hg(0) uptake fluxes on an ecosystem scale, considering
the systematic variations in crown height, needle age and tree species. Our bottom-up approach integrates
aboveground foliar Hg(0) uptake rates over the entire growing season and the whole tree level. We thus suggest
that our approach provides a robust method to assess foliar Hg(0) uptake fluxes on a species level as well as on
an ecosystem scale at a high temporal resolution. This approach is complementary to litterfall mass balances
approaches, which provide Hg deposition estimates integrated over an entire year. We suspect that the foliar Hg
uptake fluxes measured in this study represent net Hg(0) uptake fluxes as the increase of foliar Hg concentration
was linear with time which would include possible Hg(0) re-emission from foliage (Yuan et al., 2019). With the
bottom-up approach presented here, it is thus possible to obtain net foliar Hg(0) uptake fluxes that are temporally
resolved over the growing season depending on the number of temporal foliar Hg measurements. The linear uptake
of Hg(0) observed in this study across 10 European sites and for 6 different species suggests that forest foliage
take up Hg(0) from the atmosphere over the entire growing season, supporting the notion that foliar uptake of
Hg(0) drives the seasonal depletion in atmospheric Hg(0) in the Northern Hemisphere (Jiskra et al., 2018).
Our study demonstrates that foliar Hg uptake is an important deposition pathway to terrestrial surfaces and exceeds
wet deposition by a factor of 4 on average. In contrast to Hg(II) in wet deposition, which is monitored in
atmospheric deposition networks (EMEP, 2016; Pacyna et al., 2009), there is no standardized and established
program to monitor Hg deposition in foliage or litterfall across Europe. We call for including foliar mercury
deposition in monitoring networks on a country and international level. Robust and standardized data on the
development of Hg deposition to foliage and forest ecosystems will allow to assess the effectiveness of the
Minamata convention on mercury (Minamata Convention, 2019) and impact of climate change on mercury
deposition to terrestrial ecosystems in the future.

### Author contribution

M.J. designed the study. L.W. and C.J. carried out the field sampling and analytical measurements. L.W.
performed the data analysis. S.O. and G.H. gave experimental advice and sampling support. C.A. and A.K.
provided feedback and research infrastructure. L.W. wrote the manuscript in consultation with M.J. All authors
discussed the manuscript and provided comments.

### Acknowledgment

We are grateful to Fabienne Bracher, Emanuel Glauser and Judith Kobler Waldis for their help with foliage sample
preparation and analysis. We acknowledge the Swiss Canopy Crane II (SCCII) Site at Hölstein operated by the
Physiological Plant Ecology Research Group at the University of Basel and thank André Kühne and Matthias
Arend for their on-site support. We thank Frank Wania from the University of Toronto for contributing valuable
mercury passive air samplers and activated carbon. Hans-Peter Dietrich and Stephan Raspe from the Bavarian
State Institute of Forestry thankfully provided us with multi-year foliage samples from Bavaria. We acknowledge
ICOS Sweden for providing data from Hyltemossa, Norunda and Svartberget and we would like to thank Irene
Lehner, Tobias Biermann, Michal Heliasz, Antonin Kusbach, Johan Ahlgren, Ulla Nylander, Mikael Holmlund,
Pernilla Löfvenius and Per Marklund for assistance in foliage sampling and experimental support. Volkmar
Timmermann and Helge Meissner from NIBIO gratefully organized and performed foliage sampling at Hurdal.
We thank Elke Bieber, Frank Meinhardt and Rita Junek from the German Federal Environment Agency for their
support at Schauinsland and Schmücke air monitoring sites. We are grateful to Michelle Nerentorp Mastromonaco
and Ingvar Wängberg from IVL and Eva-Britt Edin for foliage sampling support and site access at Råö and

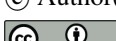



Bredkälen. We thank Katriina Kyllönen from FMI and Valtteri Hyöky for foliage sampling assistance and site
access at Pallas. Special thanks go to Jann Launer for drawing Fig. 2. Finally, we thank Christian Körner for his
helpful answers to questions on plant physiology.
**Financial support**
The work of this paper was funded by the Swiss National Science Foundation (SNSF) project 174101. The crane
at the SCCII Site is funded by the Swiss Federal Office for the Environment (FOEN). The Swedish research
infrastructures, ICOS and SITES, are both financed by the Swedish Research Council and partner universities.
**Data availability**
Foliar Hg uptake fluxes at all sites are given in the Supporting Information. Hg concentrations, metadata of all
foliage samples collected in this study are accessible at https://zenodo.org/record/3957873#.XxmttOfRabh
**Competing interests**
The authors declare that they have no conflict of interest.

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
