# Peer review of "A bottom-up quantification of foliar mercury uptake fluxes"

_Biogeosciences, 2020_

## Referee Comment (RC1) · Charles T. Driscoll (Referee) · 6 Oct 2020

The manuscript "A bottom-up quantification of foliar mercury uptake fluxes across Europe" by Wohlgemuth et al. is a detailed analysis of foliar uptake of mercury at 10 forest sites along a latitudinal gradient in Central Europe. The authors use these data to extrapolate their measurements to values of foliar mercury uptake for Europe and globally. I must say I review a lot of papers and this has be the cleanest manuscript I have ever read. My hat is off to the authors. Thank you for a very well-written, well organized and comprehensive study of foliar mercury uptake by trees including an analysis of how site level data can be used to scale up estimates of this important transfer of mercury to larger spatial scales. The authors' analysis and results are consistent with less comprehensive studies in the literature. The authors do a great job of comparing

the results with observations in the literature. I love the Methods, including figure 2. The methods are very clear. I have virtually no comments on this paper. It is well done and a pleasure to read. Just a few comments: 1. The authors use "between" when they should use "among" on lines 107, 243, 395, 403, 406, 409 and 460. 2. Page 2, line 45. . . . Earth . . . 3. Page 5, line 148. . . . dried and ground for . . . 4. Page 15, line 448. I just reviewed another paper by one of the authors of this paper that provides a global estimate of litter mercury deposition from vegetation which is an order of magnitude greater than the guesstimate provided here (1,730 – 2, 070 Mg yr-1). Given that discrepancy the authors may want to rethink their global estimate of litter mercury deposition in this paper. 5. Page 16, line 476. The authors could note that the U.S National Atmospheric Deposition program has a litter mercury network that could be cited (http://nadp.slh.wisc.edu/newissues/litterfall/). This is a terrific paper. I strongly endorse its publication. Kudos to the authors.

---

## Referee Comment (RC2) · Anonymous Referee #2 · 7 Oct 2020

Paper by Wohlgemuth et al. dealing with bottom-up quantification of foliar mercury uptake fluxes is really a notable contribution to the field of Hg foliar uptake quantification. This study deals with 10 sites located across a transect from Switzerland to northern part of Finland. Paper is well written and scientifically sound. Four species uptake rates were quantified and results of the study were up-scaled to the European and World measures. I have no major comments that would have to be addressed. But after reading, I was left with an unanswered question (mentioned by authors in Introduction) whether coniferous or deciduous trees have greater Hg concentration in their foliage. I looked for the data on Hg concentration (ng/g) in foliage at each site and I only found needle age class concentrations in Fig.SI3. I could not find relevant data for the deciduous species. . . Could Table S1 be amended with a column of Hg concentrations for all sites? Author could consider comment on differences between deciduous and coniferous trees across sites? Mentioned wet Hg(II) deposition at 5 selected sites was quite low inline with data from other European sites, could you be more specific of methods or protocols that were used at these sites. Authors postulate that the wet deposition rate covers the same period – so is it or is it not annual wet Hg(II) deposition rate?

---

## Author Response (AR1)

**Author Response to Reviewer Comments**

We would like to thank again the reviewers for their feedback and the editor for the effective handling of this manuscript. In this document we include the author responses to the two reviews and the manuscript text where all changes made during revision are highlighted in red.

**Response to Charles T. Driscoll**

The referee comments are in black while the author comments are in **bold print** and blue.

The manuscript "A bottom-up quantification of foliar mercury uptake fluxes across Europe" by Wohlgemuth et al. is a detailed analysis of foliar uptake of mercury at 10 forest sites along a latitudinal gradient in Central Europe. The authors use these data to extrapolate their measurements to values of foliar mercury uptake for Europe and globally. I must say I review a lot of papers and this has be the cleanest manuscript I have ever read. My hat is off to the authors. Thank you for a very well-written, well organized and comprehensive study of foliar mercury uptake by trees including an analysis of how site level data can be used to scale up estimates of this important transfer of mercury to larger spatial scales. The authors' analysis and results are consistent with less comprehensive studies in the literature. The authors do a great job of comparing the results with observations in the literature. I love the Methods, including figure 2. The methods are very clear. I have virtually no comments on this paper. It is well done and a pleasure to read.

**Thank you very much for this positive comment to our study. We believe that there is more research needed to refine and further quantify foliar Hg uptake fluxes in Europe and in other parts of the world. Your positive feedback highly motivates us to make an effort and reliably validate the bottom-up approach (Fig 2) on a larger spatial scale.**

Just a few comments: 1. The authors use "between" when they should use "among" on lines 07, 243, 395, 403, 406, 409 and 460. 2. Page 2, line 45. … Earth … 3. Page 5, line 148. … dried and ground for …

**Thank you, we changed all accordingly and did some grammar revisions of the manuscript.**

4. Page 15, line 448. I just reviewed another paper by one of the authors of this paper that provides a global estimate of litter mercury deposition from vegetation which is an order of magnitude greater than the guesstimate provided here (1,730 – 2, 070 Mg yr-1). Given that discrepancy the authors may want to rethink their global estimate of litter mercury deposition in this paper.

**We will certainly keep in mind the flux estimate of the current paper for assessments of global Hg fluxes. Our extrapolation of foliar Hg uptake fluxes (line 448) extends to the global land area of temperate forests only. For the tropics, higher foliar Hg concentrations and litterfall Hg fluxes had been reported, which are of an order of magnitude greater (see e.g. Teixeira et al. 2011) than the European Hg uptake flux used for the extrapolation here. Thus, for the entire global forested area we suspect the Hg litterfall flux to be bigger than the foliar Hg uptake flux reported here. The comparison of the current flux estimate for temperate forests is further complicated by the uncertainty to which extent Hg litterfall deposition fluxes may be equated with foliar Hg uptake fluxes.**

5. Page 16, line 476. The authors could note that the U.S National Atmospheric Deposition program has a litter mercury network that could be cited (http://nadp.slh.wisc.edu/newissues/litterfall/).

**Indeed we consider the litterfall mercury monitoring by the U.S. NADP a highly valuable contribution to global litterfall sampling efforts. We added the following sentence to the introduction to give credit to the network: "Hg dry deposition is usually not routinely monitored, with the Hg litterfall monitoring network of NADP being a notable exception (Risch et al., 2012, 2017)."**

This is a terrific paper. I strongly endorse its publication. Kudos to the authors.

**Author response to Referee #2**

Paper by Wohlgemuth et al. dealing with bottom-up quantification of foliar mercury uptake fluxes is really a notable contribution to the field of Hg foliar uptake quantification. This study deals with 10 sites located across a transect from Switzerland to northern part of Finland. Paper is well written and scientifically sound. Four species uptake rates were quantified and results of the study were up-scaled to the European and World measures. I have no major comments that would have to be addressed.
**We thank the referee for this positive evaluation of the paper and for the comments.**

But after reading, I was left with an unanswered question (mentioned by authors in Introduction) whether coniferous or deciduous trees have greater Hg concentration in their foliage. I looked for the data on Hg concentration (ng/g) in foliage at each site and I only found needle age class concentrations in Fig.SI3. I could not find relevant data for the deciduous species… Could Table S1 be amended with a column of Hg concentrations for all sites? Author could consider comment on differences between deciduous and coniferous trees across sites?

**The focus of this study was on flux calculation, which is why the subject of foliar Hg concentrations might have been cut short. However, we agree that it is important to clearly resolve confusion related to the difference in Hg concentration and Hg pools between deciduous and coniferous foliage. We believe that part of this confusion originates from the physiological diversity of the two tree functional groups. Coniferous needles accumulate Hg over a life cycle of multiple years but exhibit lower Hg concentrations compared to deciduous leaves of the same age. In order to visualize this discrepancy we created a table including average (± std) peak season (August) foliar Hg concentration values measured at the SCCII Forest Site Hölstein. Hölstein is a mixed forest thus allowing the sampling of various tree species side by side. Comparing Hg concentrations of various tree species growing at the same side provides the benefit of eliminating side-specific parameters impacting foliar Hg concentrations like time of sampling, Hg(0) in air (see Section 3.3) or sampling strategy (see Line 404 – 406 in Section 3.5). We thus prefer to answer the question of Hg concentration differences between tree functional groups with data from Hölstein. The average Hg concentration in beech and oak leaves in Hölstein in August is 21.7 ± 2.9 ng Hg $g_{d.w.}$ (n = 3 trees sampled at top canopy) and 22.7 ± 4.1 ng Hg $g_{d.w.}$ (n = 4 trees sampled at top canopy) respectively. The corresponding average Hg concentration of pine and spruce needles sprouted in the same season as leaves (thus same age as leaves) is lower than in leaves, being 6.5 ± 0.6 ng Hg $g_{d.w.}$ (n = 2) and 8.1 ± 2.2 ng Hg $g_{d.w.}$ (n = 3) respectively. However, multi-year old pine and spruce needles exhibit average Hg concentration values approaching the range of leaves: 13.2 ± 3.1 ng Hg $g_{d.w.}$ (one year old pine needles, n = 3), 12.8 ± 1.3 ng Hg $g_{d.w.}$ (one year old spruce needles, n = 3), 20.2 ± 5.5 ng Hg $g_{d.w.}$ (two year old spruce needles, n = 2) and 26.6 ± 7.4 ng Hg $g_{d.w.}$ (three year old spruce needles, n = 2). We included this table (Table S4, Section S8) along with an explanatory text in the Supporting Information. We expanded the main paper with the following paragraph in Section 3.1: "The continued Hg accumulation by needles over their entire life cycle has implications for the comparability of foliar Hg concentrations in needles and deciduous leaves. Deciduous leaves (beech and oak) exhibit higher average Hg concentrations than coniferous needles (pine and spruce) of the same age ($y_0$) (see Table S4 for data from Hölstein site). However, multi-year old pine and spruce needles can reach average Hg**

**concentration values higher than leaves (S8). We stress that needle age has to be reported in publications in order to avoid confusion when comparing foliar Hg concentrations of tree functional groups (deciduous vs. coniferous).” By creating a separate Hg concentration table we hope to provide more clarity on the issue of Hg concentration differences between deciduous and coniferous foliage than an expansion of Figure SI3 could. We do not further expand on the issue in Table SI1 in order to not obscure the distinction between concentrations and fluxes.**

Mentioned wet Hg(II) deposition at 5 selected sites was quite low inline with data from other European sites, could you be more specific of methods or protocols that were used at these sites.

**We included explanation on wet deposition measurement including references in the Material & Methods part of the paper (Section 2.2): “At 5 locations (Schauinsland, Schmücke, Råö, Bredkälen and Pallas) Hg(II) wet deposition measurements were performed by the operators of the research sites. At Schauinsland and Schmücke Eigenbrodt NSA 181/KD (Eigenbrodt GmbH, Königsmoor Germany) samplers were employed for collecting samples and total Hg was measured using atomic fluorescence spectroscopy (see UBA, 2004 for details on analysis). At Råö, Bredkälen and Pallas wet deposition was sampled according to EMEP protocol (NILU, 2001) (refer to Torseth et al., 2012 for an overview of EMEP)“ We do not know why Hg(II) wet deposition is lower compared to other European sites. A possible reason might be low precipitation amounts during the dry summer of 2018 (sampling period).**

Authors postulate that the wet deposition rate covers the same period – so is it or is it not annual wet Hg(II) deposition rate?

**The wet deposition rate in our paper covers the period from first to last foliage sampling event at each site respectively. To make this clear we included an explanatory parenthesis in Line 439: “ (from first to last foliage sampling event respectively)“**

[revised manuscript text omitted]